# COVID-19 pandemic-related stress and substance use behaviors among people with HIV – a mixed method analysis

Sarah L. Rossi[1,2]*, Debbie M. Cheng[3], Yuliia Sereda[4], Ve Truong[2], Jennifer J. Carroll[5], Tetiana Kiriazova[6], Sara Lodi[3], Amy Michals[7], Anita Raj[8,9], Evgeny Krupitsky[10,11], Dmitry Lioznov[10,12], Jeffrey H. Samet[2,13,14], Karsten Lunze[2]

**1** Department of Health Behavior, Gillings School of Global Public Health, University of North Carolina at Chapel Hill, Chapel Hill, North Carolina, United States of America, **2** Department of Medicine, Section of General Internal Medicine, Boston Medical Center, Clinical Addiction Research and Education (CARE) Unit, Boston, Massachusetts, United States of America, **3** Department of Biostatistics, Boston University School of Public Health, Boston, Massachusetts, United States of America, **4** Center for Evidence Synthesis in Health, Brown University School of Public Health, Providence, Rhode Island, United States of America, **5** Department of Sociology and Anthropology, North Carolina State University, Raleigh, North Carolina, United States of America, **6** Ukrainian Institute on Public Health Policy, Kyiv, Ukraine, **7** Biostatistics and Epidemiology Data Analytics Center, Boston University School of Public Health, Boston, Massachusetts, United States of America, **8** Newcomb Institute, Tulane University, New Orleans, Louisiana, United States of America, **9** Tulane School of Public Health and Tropical Medicine, New Orleans, Louisiana, United States of America, **10** Pavlov University, St. Petersburg, Russian Federation, **11** V.M. Bekhterev National Medical Research Center for Psychiatry and Neurology, St. Petersburg, Russian Federation, **12** Smorodintsev Research Institute of Influenza, St. Petersburg, Russian Federation, **13** Department of Medicine, Section of General Internal Medicine, Boston University Chobanian & Avedisian School of Medicine, Boston, Massachusetts, United States of America, **14** Department of Community Health Sciences, Boston University School of Public Health, Boston, Massachusetts, United States of America

* slrossi@unc.edu

## Abstract

### Background

Stress and financial concerns due to the COVID-19 pandemic are well documented, with mixed evidence regarding their relationship with substance use. This mixed-methods study describes the prevalence of pandemic-related stress and financial worry, and their association with changes in substance use among people with HIV with a history of injection drug use in Russia.

### Methods

We conducted a secondary analysis of survey and qualitative data collected from two trials between May 2020 and July 2021. We used multivariable logistic regression to assess associations between pandemic-related stress and financial worry, and reported changes in illicit opioids (primary), cigarettes (secondary), and alcohol (exploratory) use. The outcome was defined as any change (increase, decrease,

**Data availability statement:** The minimal data set is available at figshare via https://doi.org/10.6084/m9.figshare.31920138.

**Funding:** The LINC-II and SCRIPT trials were funded by the National Institute on Drug Abuse (NIDA) of the US National Institutes of Health (R01DA045547 and R00DA041245). https://nida.nih.gov/. DMC, JHS, and KL were supported by funding from the National Institute of Allergy and Infectious Diseases (NIAID) as part of the Providence/Boston Center for AIDS Research (CFAR) (5P30AI042853). https://www.niaid.nih.gov/research/centers-aids-research. The funders had no role in study design, data collection and analysis, decision to publish, or preparation of the manuscript.

**Competing interests:** The authors have no conflicts of interest to disclose.

or stopping) in substance use. A thematic analysis of semi-structured interviews explored experiences of stress and substance use during the pandemic.

## Results and conclusions

Among 132 survey participants, 52% reported high pandemic financial worry and 31% reported high pandemic stress. Overall, 22% reported changes in opioid use (20% decrease, 2% increase), 21% in cigarette smoking (18% decrease, 3% increase), and 15% in alcohol use (9% decrease, 6% increase). High stress was associated with reporting any change in opioid use due to the pandemic (AOR 2.98; 95%CI: 1.20, 7.40; p = 0.02). Financial worry showed a similar but imprecise association (AOR 2.45; 95%CI: 0.96, 6.21; p = 0.06). No associations were observed for cigarette or alcohol use. Qualitative findings indicated that pandemic-related stressors and disruptions in drug access may have been linked to substance use patterns for some participants, although changes in use were not a dominant theme. Most participants did not report changes in substance use; among those who did, changes were primarily decreases. Pandemic-related stress was associated with opioid use change, though direction varied. These findings suggest that stress-related responses are context-specific and may not uniformly lead to increased substance use among PWH.

## Introduction

The SARS-CoV-2 global pandemic, declared March 11, 2020, impacted stress levels differently across populations [1,2]. While the general population initially experienced stress and anxiety, more resilient individuals had no changes and some even saw mental health improvements [2]. Researchers have termed this unique pandemic stress experience "COVID Stress Syndrome," characterized by: 1) socio-economic worries; 2) contamination fears; 3) traumatic stress symptoms; 4) xenophobic concerns; and 5) compulsive reassurance seeking [3,4]. Other researchers have more generically referred to pandemic-related mental health impacts as "post-traumatic stress symptoms" or "COVID-related stress" [5].

Pre-pandemic research established stress as a risk factor for substance use and relapse [6–8]. COVID Stress Syndrome was associated with increased alcohol and substance use [9], with individuals using substances to cope with COVID-related anxiety or distress [5,10,11]. U.S. general population studies documented increased substance use during the pandemic attributed to stress [9,12–16]. However, some U.S. subgroups, including adolescents, decreased substance use despite worsening mental health [17–19]. Global pandemic substance use reports show inconsistent patterns across countries [5,20–25], and subpopulations [26]. A 21-country European survey (April-July 2020) found substance use unchanged in half of participants, with mixed increases and decreases in the remainder across alcohol, tobacco, cannabis, and illicit drugs [27].

### Substance use among people who already used substances pre-pandemic

Given stress is documented as a risk factor for substance use, examining pandemic impacts on people already using substances prior to the pandemic is crucial. Most U.S. reports showed increased substance use in this population during the pandemic, and qualitative findings highlighted mental distress from financial strain and social isolation for people who use drugs during the pandemic [13,28–31]. A U.S. study of opioid use disorder during the pandemic found that compared to prior to the pandemic, participants used drugs more frequently to replace other activities or used alone more often, while others consumed fewer or different drugs due to supply issues [32]. Peer workers in Vietnam reported increases in methamphetamine use as a coping mechanism for boredom and stress [33]. In Italy, heroin, cocaine, and MDMA use decreased during initial lockdown, but increased during re-opening, while alcohol use increased during lockdowns and remained elevated [34]. Among participants in Japan with previous illicit substance use showed correlation between higher COVID-19 fear and increased illicit substance use [35]. Georgian people who use drugs reported significant declines in all measured substances during the pandemic, including alcohol, diverted medicinal methadone, and cannabis [36]. These trends clearly vary within and across geographic regions.

### Substance use among people with HIV during the pandemic

Pandemic stress and substance use trends also varied among people with HIV (PWH). PWH represent a particularly important population for examining pandemic-related stress and substance use due to intersecting psychosocial, structural, and biological vulnerabilities. PWH experience higher baseline levels of stigma, mental health burden, and substance use [37,38], which may be exacerbated during periods of widespread disruption such as the COVID-19 pandemic [38–42]. Moreover, stress and substance use have direct implications for HIV-related outcomes, including antiretroviral therapy adherence, immune functioning, and engagement in care, making pandemic-related changes especially consequential for this population [40,43–48]. These overlapping conditions are often conceptualized as a syndemic, in which co-occurring epidemics interact to amplify health risks [49].

One U.S. study found increased alcohol use and mental health symptoms among individuals both with and without HIV, but no drug use changes [50]. Similarly, a U.S. study conducted in a primary care setting found no evidence of increased substance use or mental health problems among PWH during the pandemic [51]. Conversely, a large multi-site cohort of PWH in the U.S. showed continuous increases in the risk of moderate/high substance use disorders after pandemic onset, with increases in heroin, methamphetamine, and fentanyl use, and decreases in the use of prescription opioids and sedatives, attributed to increased isolation [52]. A Washington D.C. survey found 78% of PWH reported mental health impacts, and 56% reported financial impacts, with 15% increasing alcohol use and 8% increasing illicit drug use [53]. PWH with pre-pandemic substance use disorders increased illicit substance use and contact with other users during the pandemic [54]. Regarding stress-related changes, French PWH exposed to COVID-19 stressors showed greater distress and increased cannabis and drug use [55]. U.S. pandemic-related stress among PWH was associated with increased cigarette smoking [56]. Most pandemic studies involving PWH have been conducted in the U.S., showing varied results of both increases and no changes in substance use among PWH.

### Russian context

The Russian context provides a particularly important setting to examine these relationships. Russia has one of the fastest-growing HIV epidemics globally, largely driven by injection drug use and structural barriers such as limited access to harm reduction services and high levels of stigma [57–60]. Compared to many U.S. and Western European settings, Russia has more restricted availability of evidence-based addiction treatment, including the absence of opioid agonist therapy, and more punitive drug use policies, which may exacerbate vulnerability among PWH who use substances [58,59,61–64]. These structural differences may limit access to HIV and substance use care and reduce protective

resources during periods of crisis [62,65–68]. Pandemic-related disruptions in Russia, including economic instability, healthcare access challenges, and social marginalization, may therefore shape both stress experiences and substance use behaviors in distinct ways [69–72]. Despite these factors, there is limited research examining pandemic-related stress and substance use among PWH in Russia or similar contexts. Understanding lessons from the Russian context is vital in informing responses to future pandemics or other large-scale disruptions in settings with growing HIV epidemics.

## Objective

Given conflicting regional reports on changes in substance use, the limited representation of non-U.S. contexts, and the unique structural vulnerabilities faced by PWH in settings such as Russia, additional research is needed to understand how pandemic-related stressors influence substance use among PWH globally. This mixed-methods analysis contributes to the global pandemic impact literature by examining Russian PWH who used substances prior to the pandemic. Using data from two PWH cohorts who injected drugs before the pandemic, we examined relationships between pandemic-related stress and financial worry on changes in opioid, cigarette, and alcohol use. In-depth interviews from a subset of participants provide context for quantitative findings. This analysis offers insight into pandemic-related stress impacts on PWH with a history of drug injection. With future pandemic threats, understanding the impacts of pandemic-related stress and financial worry on diverse vulnerable populations is paramount [73].

## Methods

### Quantitative study design and participants

We performed a cross-sectional analysis of COVID-specific data collected via participant interviews between May 1, 2020, and July 31, 2021, from subsamples of two randomized controlled trials (RCTs) in St. Petersburg, Russia, LINC-II and SCRIPT: Both studies enrolled similar populations of PWH who injected drugs. Key eligibility criteria for these two trials can be found in Table 1 as well as the published descriptions of protocols and main study results [74–77]. Participants in

**Table 1. Key eligibility criteria of parent studies.**

|  | LINC-II[a]<br>(n = 225, parent study) | SCRIPT[b]<br>(n = 100, parent study) |
|---|---|---|
| **Inclusion Criteria** |  |  |
| Age 18 years or older | X | X |
| HIV-positive[c] | X | X |
| History of injection drug use (self-report) | X |  |
| Current injection drug use (past 30 days) (self-report) |  | X |
| Not currently on ART (self-report) |  | X |
| **Exclusion Criteria** |  |  |
| ART use in the past 30 days prior to hospitalization | X |  |
| Breastfeeding or being pregnant | X |  |
| Cognitive impairment precluding informed consent | X | X |
| Acute psychiatric illness | X | X |

[a]LINC-II was an RCT among PWH with a history of injection drug use that tested an intervention of rapid access to ART, receipt of naltrexone for opioid use disorder, and strength-based case management sessions over 12 months.

[b]SCRIPT was an RCT among PWH with current injection drug use and not receiving ART that tested an intervention of three group sessions of Acceptance and Commitment Therapy addressing HIV and substance use stigma over one month.

[c]HIV status determined by HIV diagnosis in medical record for LINC-II participants and self-report for SCRIPT participants.

both studies provided written consent. The **L**inking **I**nfectious and **N**arcology **C**are – Part **II** (LINC-II) study (NCT03290391) was a two-armed RCT that randomized 225 PWH receiving care in a Russian addiction hospital into an intervention that included rapid ART access, naltrexone for opioid use disorder, and strength-based case management over 12 months [74,75]. The **S**tigma **C**oping to **R**educe HIV risks and **I**mprove substance use **P**revention and **T**reatment (SCRIPT) study (NCT03695393) was a two-armed RCT that randomized 100 PWH (self-report) receiving services from a harm reduction civil society organization into an intervention that consisted of three Acceptance and Commitment Therapy group sessions addressing HIV and substance use stigma over one month [60,76].

## Qualitative study design

We analyzed qualitative data from SCRIPT participants collected during a similar timeframe (March 1, 2021 – July 31, 2021). Participants provided verbal consent which was tracked in the REDCap study management project. The original SCRIPT COVID qualitative interviews evaluated the intervention and COVID-19's impact on participant wellbeing [78,79]. The present analysis focuses on qualitative data regarding pandemic stress and substance use to contextual understanding of quantitative findings.

## Mixed methods study design

This study used a convergent mixed-methods study design, in which quantitative and qualitative data were collected during a similar time period, analyzed separately, and integrated during interpretation. This approach allows qualitative findings to provide contextual insight into quantitative results while maintaining independent analytic processes. This classification is consistent with established definitions of convergent mixed-methods designs [80]. Integration occurred at the interpretation stage, where qualitative findings were used to contextualize and explain patterns observed in the quantitative results.

## Survey data collection

The present study includes 108 LINC-II participants and 24 SCRIPT participants who completed a COVID-19 survey during one of their study visits (the 6- or 12-month study visits for LINC-II or the 6-month study visit for SCRIPT), while both trials were ongoing. Participants who had already completed their parent trial and were therefore unavailable for the COVID-19 survey, were not included in the quantitative analysis. Trained research staff administered the survey in-person or by phone due to pandemic restrictions. Relevant variables for the present analysis included substance use during the pandemic, pandemic-related financial worries and pandemic-related overall stress.

## Qualitative data collection

We conducted semi-structured interviews with 25 SCRIPT participants between March and July 2021. This sample size was determined by the scope of the parent study and was appropriate for the qualitative aims of this analysis. We used a range-maximizing sampling strategy to capture diverse perspectives and experiences across participants [81]. This approach prioritizes variation in participant characteristics and experiences, allowing for a broader understanding of the phenomenon of interest. While this strategy enhances the range of perspectives represented, it is not designed to achieve thematic saturation, as the goal is to document heterogeneity rather than exhaustively capture all themes. Given this sampling approach, thematic saturation was not the primary criterion for determining sample size.

Research team members selected participants representing both study arms, various living situations, employment statuses, genders, and ages from the parent study. Not all qualitative participants completed the COVID survey due to parent study completion timing. Remote interviews averaged 60 minutes via video conferencing or voice call. The semi-structured guide covered the following domains determined *a priori*: historical and current substance use, HIV treatment history, pandemic-related changes to drug markets and daily life, and pandemic benefits or challenges. Domains

were selected based on related literature and knowledge from the local research team. All interviews were conducted and recorded in Russian.

### Survey measures

**Outcomes.** Our outcomes were any change in use of opioids (primary), cigarettes (secondary), and alcohol (exploratory) due to the pandemic. For each substance, participants were asked "Has the coronavirus pandemic changed the amount or frequency of your [opioid/cigarette/alcohol] use?" Response options were "not changed," "increased," "decreased," "stopped due to the pandemic," "never used" (cigarettes/alcohol only), "stopped using before the pandemic," "don't know," or refused to answer. The question on opioid use did not specify type of opioid, however prior literature and qualitative findings from this study indicates that illicitly manufactured methadone is the predominant opioid used in this setting [82,83]. Descriptives of all answer options, however for the main analysis, we dichotomized into "change" and "no change." We collapsed "increased," "decreased," and "stopped" due to the pandemic into "change." Participants who responded with "never used," "stopped using before the pandemic," or "don't know" were excluded from the main analysis in order to not dilute or bias associations. We chose "change" versus "no change" outcomes (as opposed to increase versus decrease/no change) due to small sample sizes and prior literature showing inconsistent pandemic substance use changes among people with pre-pandemic substance use [32–36].

**Main independent variables.** The main independent variables were self-reported financial worry and overall stress due to the pandemic. Participants were asked "On a scale from 1 to 5, where 1 is lowest and 5 is highest, please rate your level of worry about financial problems that have resulted from the coronavirus pandemic," and "On a scale from 1 to 5, where 1 is lowest and 5 is highest, please rate your overall level of stress as a result of the coronavirus pandemic." We defined "high" as scores >= 4 based on clinical judgement that above neutral scores warrants clinical attention. Sensitivity analyses using continuous measures of stress and financial worry are described below.

**Other descriptive variables.** We reported other related COVID-19 items, including COVID-19 testing history, perceived COVID-19 infection, and household member infections.

### Statistical analysis

We computed descriptive statistics for all participants overall and by their financial worry and overall stress levels. We fit separate multivariable logistic regression models to estimate associations between pandemic-related stress and financial worry with changes in opioids (primary), cigarettes (secondary), and alcohol (exploratory) use due to the pandemic. We report adjusted odds ratios (AOR) and 95% confidence intervals (CI) from the logistic regression models. We used two-tailed tests and a significance level of 0.05 for all models. Models adjusted for gender, age, study, and randomization group as potential confounders. We controlled for study and intervention exposure using a four-category variable representing the cross-classification of study (SCRIPT vs. LINC-II) and intervention assignment (control vs. intervention). This variable accounts for potential differences in intervention exposure, as participants in intervention arms may have had greater support for coping with stress and substance use. The distribution of participants across categories was as follows: SCRIPT control (n = 5, 3.8%), SCRIPT intervention (n = 19, 14.4%), LINC-II control (n = 56, 42.4%), and LINC-II intervention (n = 52, 39.4%). We performed all analyses using SAS 9.4 software (Cary, NC, USA).

In sensitivity analyses, we examined models treating stress and financial worry as continuous predictors to assess whether associations reflected linear or threshold relationships. We also estimated models including both variables simultaneously to assess their independent associations.

### Qualitative analysis

Two Russian-English fluent team members (JC, TK), including a native Russian speaker fluent in English and a native English speaker fluent in Russian, conducted, transcribed, and coded the audio-recorded interviews in Russian. After

independently coding selected interviews openly and discussing emergent themes, we developed an English code-book with both deductive (substance use, HIV treatment access, drug market changes, stigma, police interactions) and inductive codes (finances, substance use behavior changes, family relationships, social networks, mental health). The codebook was discussed and refined by the entire study team. To ensure consistency in coding, the two coders compared initial coding decisions and resolved discrepancies through discussion, refining code definitions as needed. After code-book finalization, one coder (JC) coded all transcripts and created detailed English memos for each interview.

For the present analysis, one team member (SLR) reviewed memos to identify themes regarding finances, substance use behavior changes, stress, mental health, and drug market changes, to explain quantitative results. Regular team discussions were held to review emerging themes and ensure interpretations were grounded in the data. Representative quotes were selected from transcripts and translated by JC.

To support translation accuracy, all quoted excerpts included in this manuscript were translated from Russian to English by a bilingual team member (JC) and reviewed and approved by another bilingual team member (TK). Several other members of the study team with native or advanced fluency in both languages participated in the review, analysis, and/or translation of those interviews at various stages of the project. The analysis used MAXQDA2020 software (VERBI Software, Berlin, Germany).

### Ethics statement

Institutional Review Boards of Boston University Medical Campus and Pavlov State Medical University approved all activities. Elon University approved qualitative research activities.

Additional information regarding the ethical, cultural, and scientific considerations specific to inclusivity in global research is included in the Supporting Information (S1 Checklist).

## Results

### Quantitative survey findings

Survey respondents (n = 132) had a median age of 38 years and over a third (39.4%) were female (Table 2). Few (11.4%) survey respondents believed they had been infected with COVID-19 regardless of a COVID-19 test or diagnosis, and fewer still (4.5%) reported a positive COVID-19 test or diagnosis. Most participants (59.8%) reported not having been tested for SARS-CoV-2. Few (15.9%) reported a household member or close contact with a known COVID-19 diagnosis. Over half (52.3%) reported high financial worry and almost a third (31.1%) reported high overall stress due to the pandemic. Overall stress and financial worry due to the pandemic were moderately correlated when continuous (Spearman $\rho = 0.61$) or dichotomized (Spearman $\rho = 0.41$), indicating related but non-redundant constructs.

A majority of participants reported no change in their substance use due to the pandemic for all substances (65.9% no opioid use change, 77.3% no cigarette use change, 79.6% no alcohol use change) (Table 2). Among all survey respondents who used opioids (n = 132), 15.2% reported decreasing their use, 4.5% reported stopping their use, and 2.3% reported increasing their use due to the pandemic. Some participants reported stopping opioid use before the pandemic (12.1%); these participants were excluded from the main analysis. For survey respondents who used cigarettes (n = 128), 18.0% reported decreasing their cigarette use and 3.1% reported increasing their cigarette use. One participant reported not knowing about their cigarette use and one participant reported that they had never used cigarettes; these participants were not included in the main analysis. For survey respondents who used alcohol (n = 103), 5.8% reported decreasing drinking, 2.9% reported stopping drinking, and 5.8% reported increasing drinking due to the pandemic. Five percent stopped drinking before the pandemic and one person did not know if their alcohol use changed; these participants were excluded from the main analysis. After dichotomizing, 25.0%, 21.4%, and 15.5% of people respectively using opioids, cigarettes, and alcohol, reported changing their use in some way due to the pandemic.

**Table 2. Characteristics of PWH with a history of injection drug use (n = 132).**

|  | n | % |
|---|---|---|
| LINC-II participants | 108 | 81.8% |
| SCRIPT participants | 24 | 18.2% |
| Female (n, %) | 52 | 39.4% |
| Median age (25th, 75th), years | 38.0 (35.0, 40.0) | |
| COVID-19 testing/diagnosis (self-reported) | | |
| Negative (n, %) | 47 | 35.6% |
| Positive (n, %) | 6 | 4.5% |
| No test/diagnosis (n, %) | 79 | 59.8% |
| Believe you were infected with COVID-19 | | |
| No (n, %) | 117 | 88.6% |
| Yes (n, %) | 15 | 11.4% |
| Household member or close contact infected with COVID-19 | | |
| No (n, %) | 111 | 84.1% |
| Yes (n, %) | 21 | 15.9% |
| Financial worry due to the pandemic (n,%) | | |
| 1 (Not at all) | 20 | 15.2% |
| 2 | 15 | 11.4% |
| 3 | 28 | 21.2% |
| 4 | 22 | 16.7% |
| 5 (Very much) | 47 | 35.6% |
| High financial worry due to the pandemic (n, %)[a] | 69 | 52.3% |
| Overall stress due to the pandemic (n,%) | | |
| 1 (Not at all) | 26 | 19.7% |
| 2 | 26 | 19.7% |
| 3 | 39 | 29.5% |
| 4 | 24 | 18.2% |
| 5 (Very much) | 17 | 12.9% |
| High overall stress due to the pandemic (n, %)[a] | 41 | 31.1% |
| Opioid use (n = 132) (n, %) | | |
| No change due to the pandemic | 87 | 65.9% |
| Decreased due to the pandemic | 20 | 15.2% |
| Stopped due to the pandemic | 6 | 4.5% |
| Increased due to the pandemic | 3 | 2.3% |
| Stopped using opioids before the pandemic | 16 | 12.1% |
| Opioid use change (dichotomized) due to the pandemic (n = 116)[b] | | |
| No change | 87 | 75.0% |
| Any change | 29 | 25.0% |
| Cigarette use (n = 128) (n, %) | | |
| No change due to the pandemic | 99 | 77.3% |
| Decreased due to the pandemic | 23 | 18.0% |
| Increased due to the pandemic | 4 | 3.1% |
| Never used | 1 | 0.8% |
| Don't know | 1 | 0.8% |
| Cigarette use change (dichotomized) due to the pandemic (n = 126)[b] | | |
| No change | 99 | 78.6% |

*(Continued)*

**Table 2.** (Continued)

| | n | % |
|---|---|---|
| Any change | 27 | 21.4% |
| Alcohol use change due to the pandemic (n = 103) (n, %) | | |
| No change due to the pandemic | 82 | 79.6% |
| Decreased due to the pandemic | 6 | 5.8% |
| Stopped drinking due to pandemic | 3 | 2.9% |
| Increased due to the pandemic | 6 | 5.8% |
| Stopped drinking before the pandemic | 5 | 4.9% |
| Don't know | 1 | 1.0% |
| Alcohol use change (dichotomized) due to the pandemic (n = 97)[b] | | |
| No change | 82 | 84.5% |
| Any change | 15 | 15.5% |

[a]Financial worry and overall stress were dichotomized for analyses. Participants endorsing a 4 or 5 were classified as high financial worry or high overall stress.

[b]Change was dichotomized as "any change" (decreased, increased, or stopped due to the pandemic) versus "no change." Participants reporting never used, stopping prior to the pandemic, or don't know were excluded from analyses.

We observed the following outcomes for the groups with high versus low financial worry: change in their reported opioid use due to pandemic (Table 3): 32.3% (20/62) versus 16.7% (9/54); change in their reported cigarette use due to pandemic (Table 4): 21.2% (14/66) versus 21.7% (13/60); change in their reported alcohol use due to the pandemic (Table 5): 16.0% (8/50) versus 14.9% (7/47). We observed the following outcomes for the groups with high versus low overall stress—change in their reported opioid use due to the pandemic (Table 3): 40.5% (15/37) versus 17.7% (14/79); change in their reported cigarette use due to the pandemic (Table 4): 22.0% (9/41) versus 21.2% (18/85); change in their reported alcohol use due to the pandemic (Table 5): 16.7% (5/30) versus 14.9% (10/67).

In multivariable logistic regression analyses, participants reporting high overall stress due to the pandemic had more than twice the odds of reporting any change in opioid use compared to those reporting low stress (Table 3), which was statistically significant (adjusted odds ratio [AOR] 2.98; 95% CI: 1.20, 7.40; p = 0.02). Participants with high financial worry also had higher odds of reporting any change in opioid use due to the pandemic (AOR 2.39; 95% CI: 0.97, 5.85; p = 0.057), although the confidence interval included the null, indicating imprecision in the estimate. As noted in descriptive analyses, most reported changes in opioid use reflected decreases rather than increases. We did not detect associations between high financial worry and high overall stress and changes in cigarette smoking or alcohol consumption due to the pandemic (Tables 4 and 5).

In post-hoc sensitivity models including both financial worry and overall stress simultaneously, effect estimates were attenuated and no longer statistically significant, likely reflecting shared variance and limited precision (Table 6). However, results were consistent in direction with the primary analysis for opioid use. When modeled as continuous predictors, both financial worry (AOR = 1.51; 95% CI: 1.06, 2.15; p = 0.02) and overall stress (AOR = 1.83; 95% CI: 1.24, 2.69; p < 0.01) were significantly associated with opioid use change, supporting a graded relationship and indicating that findings were not dependent on dichotomization (Table 7).

## Qualitative findings

We interviewed 25 SCRIPT participants for the qualitative sub-study, with ages ranging from 20 to 56 years, and 48% of the sample were women. Further interviewee characteristics are described elsewhere [78,79]. Below we qualitatively

**Table 3. Association between opioid use change and financial worry and stress (n = 116).**

| | Opioid use change[a] (25.0%, n = 29) | AOR (95% CI) | p-value |
|---|---|---|---|
| COVID-19 overall financial worry[b] | | | |
| **High, n = 62** | 32.3% (n = 20) | 2.45 (0.96, 6.21) | 0.06 |
| **Low, n = 54** | 16.7% (n = 9) | Ref | |
| COVID-19 overall stress[b] | | | |
| **High, n = 37** | 40.5% (n = 15) | **2.98 (1.20, 7.40)** | **0.02** |
| **Low, n = 79** | 17.7% (n = 14) | Ref | |

AOR = adjusted odds ratio; CI = confidence interval

[a]Change defined as increased, decreased, or stopped opioid use due to the pandemic; participants reporting pre-pandemic cessation were excluded.

[b]High = score 4–5; Low = score 1–3. Models adjusted for gender, age, study, and intervention group.

**Table 4. Association between cigarette use change and financial worry and stress (n = 126).**

| | Cigarette use change[a] (21.4%, n = 27) | AOR (95% CI) | p-value |
|---|---|---|---|
| COVID-19 overall financial worry[b] | | | |
| **High, n = 66** | 21.2% (n = 14) | 1.00 (0.42, 2.38) | 0.99 |
| **Low, n = 60** | 21.7% (n = 13) | Ref | |
| COVID-19 overall stress[b] | | | |
| **High, n = 41** | 22.0% (n = 9) | 1.10 (0.43, 2.81) | 0.84 |
| **Low, n = 85** | 21.2% (n = 18) | Ref | |

AOR = adjusted odds ratio; CI = confidence interval

[a]Change defined as increased or decreased cigarette use due to the pandemic; participants reporting never used or don't know were excluded.

[b]High = score 4–5; Low = score 1–3. Models adjusted for gender, age, study, and intervention group.

**Table 5. Association between alcohol use change and financial worry and stress (n = 97).**

| | Alcohol use change[a] (15.5%, n = 15) | AOR (95% CI) | *p-value* |
|---|---|---|---|
| *COVID-19 overall financial worry[b]* | | | |
| **High, n = 50** | 16.0% (n = 8) | 1.14 (0.36, 3.61) | 0.82 |
| **Low, n = 47** | 14.9% (n = 7) | Ref | |
| *COVID-19 overall stress[b]* | | | |
| **High, n = 30** | 16.7% (n = 5) | 0.96 (0.27, 3.35) | 0.95 |
| **Low, n = 67** | 14.9% (n = 10) | Ref | |

AOR = adjusted odds ratio; CI = confidence interval

[a]Change defined as increased, decreased, or stopped alcohol use due to the pandemic; participants reporting pre-pandemic cessation or don't know were excluded.

[b]High = score 4–5; Low = score 1–3. Models adjusted for gender, age, study, and intervention group.

describe experiences from these PWH who inject drugs and changes in their 1) financial concerns, 2) emotional and psychological wellbeing, and 3) substance use and drug availability during the pandemic.

**Financial concerns during the pandemic.** Multiple interviewees shared that the biggest impact of the pandemic on their daily life was relating to financial challenges. Many had either lost their job or were unable to get a job due to the pandemic, with one interviewee explaining, *"it has become more difficult to get a job, because, due to the pandemic, lots of organizations shut down. Lots of people were let go."* A few interviewees added that the difficulty with getting a job

**Table 6. Joint adjusted model of financial worry and overall stress and substance use.**

| | Opioid use change[a] AOR (95% CI) | p-value | Cigarette use change[a] AOR (95% CI) | p-value | Alcohol use change[a] AOR (95% CI) | p-value |
|---|---|---|---|---|---|---|
| Financial worry (high vs low)[b] | 1.73 (0.62, 4.82) | 0.30 | 0.96 (0.37, 2.49) | 0.93 | 1.26 (0.31, 5.18) | 0.27 |
| Overall stress (high vs low)[b] | 2.37 (0.87, 6.43) | 0.09 | 1.12 (0.40, 0.83) | 0.83 | 0.83 (0.18, 3.78) | 0.81 |

AOR = adjusted odds ratio; CI = confidence interval

[a]Change defined as increased, decreased, or stopped use due to the pandemic; participants reporting pre-pandemic cessation, never used, or don't know were excluded.

[b]High = score 4–5; Low = score 1–3. Models adjusted for gender, age, study, and intervention group

**Table 7. Continuous model of financial worry and overall stress and substance use.**

| | Opioid use change[a] AOR (95% CI) | p-value | Cigarette use change[a] AOR (95% CI) | p-value | Alcohol use change[a] AOR (95% CI) | p-value |
|---|---|---|---|---|---|---|
| Financial worry (continuous)[b] | **1.51 (1.06, 2.15)** | **0.02** | 1.17 (0.85, 1.59) | 0.34 | 1.25 (0.82, 1.91) | 0.30 |
| Overall stress (continuous)[b] | **1.83 (1.24, 2.69)** | **<0.01** | 1.28 (0.90, 1.83) | 0.17 | 1.46 (0.92, 2.33) | 0.11 |

AOR = adjusted odds ratio; CI = confidence interval

[a]Change defined as increased, decreased, or stopped use due to the pandemic; participants reporting pre-pandemic cessation, never used, or don't know were excluded.

[b]High = score 5; Low = score 1. Models adjusted for gender, age, study, and intervention group

during the pandemic was compounded by their substance use, as they were unable to keep a regular schedule and fired *"for being late,"* or their employer would *"never tolerate an addict who was using [drugs]."*

A few interviewees explained how their financial struggles due to the pandemic worried them. One interviewee shared that he was already worried about his job due to his substance use, and that the pandemic heightened this worry. He shared, *"Like, my [current] employer knew that I had problems with drugs before, but he hired me [anyway]. And when the pandemic began, I was already starting to worry that something could happen, that I could lose this job, because it pays well."* Another shared that the source of his emotional and psychological stress was associated with financial deficits as his income decreased by half.

In order to cope with financial challenges, interviewees responded in various ways. One interviewee reported needing to steal food to survive, and that due to the pandemic, *"it's harder [to get away with such things], because there are fewer people [out on the street],"* suggesting that it's more difficult to be inconspicuous. Others shared that they were living off their savings, or relied on family support, including living off of parents' pensions or incomes with one interviewee sharing, *"It's mostly my mom who supports me. I have other folks who help as much as they can."*

Financial impacts appeared to be the primary stressor due to the pandemic, as even interviewees who reported no other changes in any other aspect of their life, shared that the loss of employment impacted them. One interviewee shared, *"The pandemic doesn't worry me, it doesn't bother me. Cause it didn't affect me so much. There were some problems with work, but that's it."* A few interviewees were able to continue their work as normal during the pandemic such as a taxi driver who said, *"everything is exactly the same, nothing's changed,"* but many did not have this option.

**Emotional and psychological wellbeing during the pandemic.** Emotional and psychological wellbeing during the pandemic among these interviewees was mixed. Some interviewees shared there *"nothing has changed"* in their life during the pandemic and did not report any psychological distress. When asked how the pandemic has impacted her, one interviewee shared, *"I don't think much of anything about it,"* and another interviewee noted that he *"[has] not seen*

*anything that would cause people to panic.*" Of those who did report psychological stress, many related this distress specifically to financial concerns as noted above.

However, a few interviewees did suggest their stress was due to the pandemic itself, not relating to finances. Several interviewees shared that they had *"feared for [their] health,"* and were worried for their friends and relatives becoming infected with COVID, especially if their relatives were elderly. One interviewee shared that everyone panicked early in the pandemic. He added that, *"I'll be honest when I say that it was the uncertainty [that affected me]. Because, most of all there was the fear for my health. Because if a lot of healthy people are losing their lives [from COVID], then those of us here with our problem…,"* referring to his HIV status and substance use. One interviewee stated that the source of his worries was that *"we [people with HIV] will be the last to be vaccinated."* Another interviewee shared that they were really worried in general at the start of the pandemic, and that their depression had increased due to social isolation. When asked about his emotional experience of losing relatives due to COVID, a younger interviewee (aged 20) shared the emotional difficulty of losing friends to both substance use and COVID:

> *"Well, we got worried every time that I found out someone close to me was sick with the coronavirus. It felt like you didn't know what was gonna happen, and also that, like, what if there were complications, because it's really such a serious disease. It was really hard for me. Not only are a lot of people who use drugs dying, even my friends, but now also because of the coronavirus I'm losing a lot of friends."*

Another interviewee shared that there was a general feeling of people being suspicious of each other, especially on public transit. He shared that, *"It's obvious that when you're out and about, when you're sitting there on the bus, and someone sitting next to you is coughing and sneezing, you know, any person will start to have all sorts of thoughts creeping into their head,"* as well as worried about what others would think of him if he started coughing.

**Drug market changes during the pandemic possibly leading to changes in substance use.** There were inconsistent reports regarding the availability, price, and quality of various illicit drug products during the pandemic. Many interviewees reported there were no changes on any of these aspects of the drug markets, including a few who mentioned that the quality of drugs had already deteriorated before the pandemic and the quality of methadone was *"just as lousy as it ever was."*

Many interviewees used "dead drops" or places where a dealer would hide the online-purchased drugs in a specific location, and the buyer would pick up their purchase at the location. This method was used prior to the pandemic as well but seems to have been in more use during the pandemic, as there was no need for handing off drugs between people, which helped mitigate COVID transmission risk. One interviewee shared that *"well, in St. Petersburg [the exchange] is now mainly through dead drops, but there are places where it's hand to hand,"* suggesting that there may be regional differences in methods of drug exchange.

Even with many interviewees utilizing these methods, others reported that there were more difficulties in accessing drugs, as the dead drops were "*a long ways away*" from their house, or dead drops were riskier with police *"on guard"* more to enforce mask restrictions. One interviewee shared that because of higher risk of being caught when using dead drops during the pandemic, she said that *"it was easier for me to not use; I thought my freedom was worth more than whatever pleasure [came from using]."* Some shared that in the beginning of the pandemic, *"they closed the borders,"* and drugs were less available, but that this went back to normal after initial restrictions were lifted. Quality of the drugs was reported as unchanged due to the pandemic by some interviewees and *"became worse"* from other interviewees.

Regarding interviewees' own substance use, many reported *"nothing has changed"* or that the pandemic *"did not affect [their drug use] in any way."* However, a few interviewees did report wanting to reduce their substance use or *"quit drugs altogether"* for various reasons including lack of money, which *"was attributed to fewer work opportunities during the pandemic. Another interviewee reported that their use decreased *"as soon as the pandemic began…"* due to, *"there came a

*time when it was difficult [to access drugs],"* but that this difficulty *"lasted maybe a week, not long."* Interviewees shared that their main substance of choice was "*methadone,*" however some shared that *"salts"* a term generally, but not exclusively, used to refer to a variety of cathinone products, "*are generally popular, their popularization is increasing.*"

In summary, interviewees shared that financial concerns due to loss of employment were a primary concern as well as some interviewees very worried about COVID itself. However, others reported that there were no changes in their life at all during the pandemic, including in terms of their substance use. However, the drug supply did affect cost of and access to drugs which some participants described as resulting in their reducing or eliminating drug use.

## Discussion

Among this sample of PWH who reported current or past injection drug use in St. Petersburg, Russia, many experienced high financial worry and moderate overall stress from the pandemic, yet most reported no change in substance use. Higher pandemic-related stress was significantly associated with reporting any change in opioid use, with descriptive findings indicating that most of these changes were reductions rather than increases. Financial worry showed a similar pattern of association with opioid use change, although estimates were imprecise. No associations were observed between stress or financial worry and changes in cigarette smoking or alcohol use.

The qualitative findings provide a nuanced understanding of the circumstances of use in the context of a pandemic. Interviewees reported varied pandemic impacts; some experienced no life changes while others faced financial stress and fear of illness. Substance use patterns were mixed, with most reporting no change and some reporting decreased use due to reduced drug access and supply disruptions. Notably, no interviewees reported increased substance use, consistent with the quantitative findings.

The observed association between stress and reporting any change in opioid use warrants careful interpretation. Although stress was associated with change in opioid use, only 25% of survey respondents reported any change in opioid use, and among those, approximately 90% reported decreases while 10% reported increases. Qualitative findings suggest that pandemic-related stressors (notably fear of COVID-19 infection) alongside disruptions in access to substances such as methadone may help explain the pattern. Financial worry was commonly described as a stressor regardless of substance use behavior in qualitative data, which may contribute to the weaker and more imprecise quantitative association observed for financial worry and any opioid use change.

### Comparison with other studies

Our findings align with some European studies, yet contrast with many U.S. reports showing increased substance use during the pandemic [12,13,30,84]. While most U.S. studies found increases in substance use due to pandemic stressors like isolation and economic difficulties, loneliness, and general anxiety among both general populations [9,14–16] and PWH [55,56], two U.S. studies among PWH found no changes in substance use [50,51], consistent with our results. Our findings may reflect initial pandemic restrictions, similar to a systematic review that found substance use decreased before sharply increasing once restrictions lifted, especially among people who already used substances [85]. However, 75–85% of our respondents reported no change across substances, aligning more with European than U.S. patterns.

European reports show more varied substance use changes than U.S. studies. A 21-country European survey found that half of participants did not change their substance use, with mixed increases and decreases among the remainder [27]. Russian participants indicated no significant changes in alcohol, tobacco, cannabis, or other illegal drugs [27], consistent with our results. In this study, participants who changed use primarily decreased consumption, similar to findings from Georgia, which may have drug markets comparable to Russia [36]. Georgian people who used drugs consumed fewer substances during April-September 2020 and shifted preferred drugs based on availability [36], potentially similar to our qualitative findings of increased use of synthetic cathinones (salts).

Importantly, our sample differs substantially from general population surveys conducted in Europe, as it consists of people with HIV with a history of injection drug use in a context marked by structural vulnerability, stigma, and constrained access to harm reduction services. Furthermore, other work has shown Russia to have very low trust in vaccines among both the population and healthcare providers, which influenced the COVID-19 response [86–88]. This likely contributed to delayed vaccine uptake and a more prolonged and severe pandemic environment, potentially exacerbating overall stress. These factors may shape both stress exposure and substance use responses, limiting direct comparability with broader population-based findings.

## Future pandemic preparedness

These results point to an important lesson for pandemic preparedness: responses must identify and address the specific stressors that shape behaviors, which may vary by context and population. Involving communities in the design of preparedness strategies can help clarify these dynamics and ensure interventions are tailored to local realities. The impact of pandemic stress on HIV was not a prominent theme in our data and thus highlights the importance of working with affected populations directly to identify their pandemic-related stressors and needs, rather than assuming their stressors and needs. Future preparedness efforts should therefore integrate stress-responsive supports into broader public health programming to reduce inequities and strengthen resilience in the face of future pandemics.

## Key findings

This mixed methods analysis describes that among Russian PWH who injected drugs prior to the pandemic, greater overall pandemic-related stress was associated with reporting any change in opioid use, with qualitative findings pointing to COVID-related fears and reduced access to substances as potential contributing factors. Importantly, most survey respondents did not report changes in substance use, and among those who did, changes were primarily decreases. These findings highlight that the relationship between stress and substance use during the pandemic may vary across settings and populations. While some studies in U.S. populations have documented increases in substance use during the pandemic, evidence from European contexts has been more heterogeneous. Together, our results suggest that pandemic-related stress does not uniformly lead to increased substance use and may instead be associated with a range of behavioral responses among PWH.

## Limitations

Internal validity was limited by year-long data collection during rapidly changing pandemic conditions preventing temporal attributions, including due to government-enforced restrictions, variants, vaccine availability. While we specified stress and substance use changes "due to the pandemic," other concurrent factors may have influenced survey respondents' experiences [89]. Self-reported data are subject to recall and social desirability biases. Further, there is potential selection bias, as only participants still engaged with follow-up visits of the parent studies completed the COVID survey. External validity was limited due to data collection solely in St. Petersburg, Russia, among a specific population. The small, non-generalizable sample lacked power to detect significant associations in this exploratory study. Qualitative interviewees (SCRIPT) differed from the full quantitative sample (LINC-II and SCRIPT combined), potentially limiting representativeness. In addition, qualitative data was analyzed from in-depth English memos, and then quotes were extracted from Russian transcripts, as opposed to direct analysis from transcripts, potentially leading to confirmation bias from the researchers. However, the study team consisted of both native Russian speakers fluent in English and native English speakers fluent in Russian, who participated in the review, analysis, and/or translation of the excerpts used in this analysis.

Due to few participants reporting increased substance use, we analyzed our outcome as "change versus no change," rather than "increased versus decreased/no change," limiting interpretation. This approach treats increase and decrease

equally, requiring careful contextualization of regression results rather than direct interpretation. However, these outcomes remain meaningful as they reflect the reality that few participants changed their substance use due to the pandemic. Type of opioids was not specified in the survey item, however all participants who specified their drug of choice in the qualitative interviews reported using illicitly manufactured methadone. This is consistent with prior literature documenting the predominance of illicitly manufactured methadone in Russia [82,83].

## Conclusions

In this sample of PWH in St. Petersburg with a history of injection drug use, many participants reported high financial worry and moderate stress due to the pandemic. Most participants did not report changes in substance use; among those who did, changes were primarily decreases. Higher overall pandemic-related stress was associated with reporting any change in opioid use. Qualitative findings suggest that pandemic-related disruptions in drug supply and access may have contributed to these patterns for some participants. These findings underscore that responses to stress during pandemics or similar complex crises are not uniform. Future interventions addressing stress and substance use should account for community-specific contexts and responses rather than assuming consistent behavioral patterns.

## Supporting information

**S1 Checklist.** Authors' responses to the PLOS ONE Inclusivity in Global Research Guidelines.Checklist detailing how the study aligns with the journal's Inclusivity in Global Research framework.
(DOCX)

## Author contributions

**Conceptualization:** Sarah L. Rossi, Karsten Lunze.

**Data curation:** Yuliia Sereda, Jennifer J. Carroll, Tetiana Kiriazova.

**Formal analysis:** Sarah L. Rossi, Jennifer J. Carroll, Tetiana Kiriazova, Amy Michals.

**Funding acquisition:** Evgeny Krupitsky, Dmitry Lioznov, Jeffrey H. Samet, Karsten Lunze.

**Investigation:** Jennifer J. Carroll, Tetiana Kiriazova.

**Methodology:** Debbie M. Cheng, Yuliia Sereda, Sara Lodi, Anita Raj.

**Project administration:** Sarah L. Rossi, Ve Truong.

**Software:** Amy Michals.

**Supervision:** Debbie M. Cheng, Evgeny Krupitsky, Dmitry Lioznov, Jeffrey H. Samet, Karsten Lunze.

**Writing – original draft:** Sarah L. Rossi.

**Writing – review & editing:** Debbie M. Cheng, Yuliia Sereda, Ve Truong, Jennifer J. Carroll, Tetiana Kiriazova, Sara Lodi, Amy Michals, Anita Raj, Evgeny Krupitsky, Dmitry Lioznov, Jeffrey H. Samet, Karsten Lunze.

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
