## [Decision Letter · Decision Letter 0]

26 Jan 2026

PONE-D-25-55857COVID-19 pandemic-related stress and substance use behaviors among people with HIV – a mixed method analysisPLOS One

Dear Dr. Rossi,

Thank you for submitting your manuscript to PLOS ONE. After careful consideration, we feel that it has merit but does not fully meet PLOS ONE’s publication criteria as it currently stands. Therefore, we invite you to submit a revised version of the manuscript that addresses the points raised during the review process.

Please note that we have only been able to secure a single reviewer to assess your manuscript. We are issuing a decision on your manuscript at this point to prevent further delays in the evaluation of your manuscript. Please be aware that the editor who handles your revised manuscript might find it necessary to invite additional reviewers to assess this work once the revised manuscript is submitted. However, we will aim to proceed on the basis of this single review if possible.  The reviewer's comments are below. Could you please revise the manuscript to carefully address the concerns raised?

We look forward to receiving your revised manuscript.

Kind regards,

Steve Zimmerman, PhD

Senior Editor, PLOS One

Journal Requirements:

4. Thank you for uploading your study's underlying data set. Unfortunately, the repository you have noted in your Data Availability statement does not qualify as an acceptable data repository according to PLOS's standards.

6. We note that there is identifying data in the Supporting Information file < SCRIPT Combined IRB Docs and LINC-II IRB docs + translations>. Due to the inclusion of these potentially identifying data, we have removed this file from your file inventory. Prior to sharing human research participant data, authors should consult with an ethics committee to ensure data are shared in accordance with participant consent and all applicable local laws.

-Location data

Please remove or anonymize all personal information (Name, Address and Dates), ensure that the data shared are in accordance with participant consent, and re-upload a fully anonymized data set. Please note that spreadsheet columns with personal information must be removed and not hidden as all hidden columns will appear in the published file.

Reviewers' comments:

Reviewer's Responses to Questions

**Comments to the Author**

1. Is the manuscript technically sound, and do the data support the conclusions?

Reviewer #1: Partly

2. Has the statistical analysis been performed appropriately and rigorously? 

Reviewer #1: I Don't Know

3. Have the authors made all data underlying the findings in their manuscript fully available?

Reviewer #1: Yes

4. Is the manuscript presented in an intelligible fashion and written in standard English?

Reviewer #1: Yes

5. Review Comments to the Author

Reviewer #1: This is a timely mixed-methods secondary analysis examining whether COVID-19–related stress and financial worry were associated with changes in opioid, cigarette, and alcohol use among people with HIV (PWH) with prior or current injection drug use in St. Petersburg, Russia, using data collected between May 2020 and July 2021. Key strengths include the focus on an under-studied, high-risk population outside the U.S. and the use of both survey and interview data to contextualize findings. With revisions—particularly clarifying measurement, outcome specification, statistical robustness, and interpretation—the paper could make a solid contribution.

Introduction

The introduction effectively highlights inconsistencies in prior findings. It would benefit from clearer justification of why PLWH are a particularly relevant population for studying pandemic stress and substance use, and how the Russian context may differ from other regions in shaping these experiences. Expanding on these points would better articulate the study’s contribution and research gap.

Methods, Results, and Discussion

Some details from the parent trials that are not directly analyzed here could be streamlined to avoid confusion (e.g., ART status, COVID infection history). The inclusion/exclusion table is useful, but the text should focus more tightly on variables relevant to this analysis.

The definition of “no change” combines “not changed,” “never used,” and “stopped using before the pandemic,” while “changed” includes increased, decreased, or stopped use due to the pandemic. Including never-users and pre-pandemic stoppers in the reference group may dilute or bias associations, particularly for alcohol and cigarettes, and creates inconsistencies across substances. If reanalysis is not feasible, a sensitivity analysis excluding these groups—or clearer discussion of this limitation—is recommended.

Causal language should be tightened. This is a cross-sectional analysis with stress and perceived change measured at the same time point, so directionality cannot be established. The manuscript should more clearly frame findings as associations and avoid implying causation.

Dichotomizing stress and financial worry at ≥4 on a 1–5 scale requires stronger justification or sensitivity analyses. Reporting correlations between stress and financial worry, or considering them jointly in models, would strengthen interpretation.

Interpretation of the main finding could be clearer. The outcome reflects “any change,” not direction, and most changes were decreases. Reframing conclusions accordingly would better align with the data. Language describing financial worry as “clinically meaningful” should be softened given imprecision in the estimate.

The mixed-methods approach appears more explanatory than truly convergent, as qualitative findings are used primarily for contextualization. Clarifying the design and noting reliance on English-language memos (rather than full transcript coding by the integrative analyst) as a limitation would improve transparency.

Selection bias is also possible, as only participants still engaged in follow-up completed the COVID survey. If available, describing differences between respondents and non-respondents—or acknowledging likely bias toward more stable participants—would be helpful.

Other observations

There is a time window inconsistency: the Abstract reports May 2020–March 2021, while the Methods reference interviews through July 2021. Please ensure consistent reporting.

The term “opioids” appears in qualitative narratives to include illicit opioids and methadone (and possibly diverted methadone). If possible, clarify what the opioid survey item specifically captured.

In Table 2, age is described as a mean in the text but presented as a median in the table; please make this consistent.

The four-category intervention variable combining study and arm should be clearly defined in the Methods, ideally with sample sizes per category.

Additional detail on qualitative rigor (e.g., whether saturation was assessed and how translation accuracy was ensured) would strengthen the qualitative methods.

While the Discussion appropriately emphasizes alignment with European findings, it would also be helpful to reiterate that this population—PWH with injection drug use in Russia—differs meaningfully from general population European surveys.

6. PLOS authors have the option to publish the peer review history of their article (what does this mean?). If published, this will include your full peer review and any attached files.

Reviewer #1: No

---

## [Author Response · Author response to Decision Letter 1]

7 Apr 2026

Formatted response to reviewer and editor comments is included in the attached files. The same text is also included below:

Dear Dr. Zimmerman,

Thank you for the opportunity to revise our manuscript entitled “COVID-19 pandemic-related stress and substance use behaviors among people with HIV – a mixed method analysis.” We appreciate the thoughtful and constructive feedback from you and the reviewer. We have revised the manuscript accordingly and believe these changes have strengthened the clarity and rigor of the paper. Below, we provide a detailed, point-by-point response to all comments. Changes in the manuscript are in tracked changes, and page numbers refer to the revised manuscript.

On behalf of all co-authors,

Sarah Rossi

Editor’s Comments

Editor comment 1: Please ensure that your manuscript meets PLOS ONE's style requirements, including those for file naming. The PLOS ONE style templates can be found at

RESPONSE: We have updated the formatting throughout the manuscript to align with PLOS ONE’s style requirements.

Editor comment 2: Please include a complete copy of PLOS’ questionnaire on inclusivity in global research in your revised manuscript. Our policy for research in this area aims to improve transparency in the reporting of research performed outside of researchers’ own country or community. The policy applies to researchers who have travelled to a different country to conduct research, research with Indigenous populations or their lands, and research on cultural artefacts. The questionnaire can also be requested at the journal’s discretion for any other submissions, even if these conditions are not met. Please find more information on the policy and a link to download a blank copy of the questionnaire here: https://journals.plos.org/plosone/s/best-practices-in-research-reporting. Please upload a completed version of your questionnaire as Supporting Information when you resubmit your manuscript.

RESPONSE: We have uploaded a copy of PLOS’ questionnaire on inclusivity in global research and referenced it in the ethics statement in the methods section as a Supporting Information file.

Changes in manuscript: This questionnaire is referenced in the ethics statement on lines 310-312 and the caption to the supporting file is located on lines 926-928.

Editor comment 3: We note that the grant information you provided in the ‘Funding Information’ and ‘Financial Disclosure’ sections do not match.

RESPONSE: We have provided updated Financial Disclosure information in the revised cover letter to match the Funding Information section.

Editor comment 4: Thank you for uploading your study's underlying data set. Unfortunately, the repository you have noted in your Data Availability statement does not qualify as an acceptable data repository according to PLOS's standards.

RESPONSE: We have uploaded our minimal dataset to figshare. Please find the repository link here: https://doi.org/10.6084/m9.figshare.31920138

Editor comment 5: Please include captions for your Supporting Information files at the end of your manuscript, and update any in-text citations to match accordingly. Please see our Supporting Information guidelines for more information: http://journals.plos.org/plosone/s/supporting-information.

RESPONSE: We have added captions for our Supporting Information file at the end of the manuscript.

Changes in manuscript: The caption for the supporting information file is on page 42, lines 925-928.

Editor comment 6: We note that there is identifying data in the Supporting Information file < SCRIPT Combined IRB Docs and LINC-II IRB docs + translations>. Due to the inclusion of these potentially identifying data, we have removed this file from your file inventory. Prior to sharing human research participant data, authors should consult with an ethics committee to ensure data are shared in accordance with participant consent and all applicable local laws.

-Location data

Please remove or anonymize all personal information (Name, Address and Dates), ensure that the data shared are in accordance with participant consent, and re-upload a fully anonymized data set. Please note that spreadsheet columns with personal information must be removed and not hidden as all hidden columns will appear in the published file.

RESPONSE: Thank you for highlighting this issue. After review, we have removed the Supporting Information file of qualitative quotes due to the sensitive nature of the qualitative data, the small sample size, and the potential risk of participant identification. In accordance with PLOS ONE’s data availability policy, we have made a de-identified quantitative dataset available via figshare (https://doi.org/10.6084/m9.figshare.31920138). Qualitative excerpts included in the manuscript have been carefully selected and reviewed to ensure that they do not contain identifying information.

Editor comment 7: If the reviewer comments include a recommendation to cite specific previously published works, please review and evaluate these publications to determine whether they are relevant and should be cited. There is no requirement to cite these works unless the editor has indicated otherwise.

RESPONSE: Thank you for this clarification. The reviewer did not recommend citing specific previously published works.

Reviewers' comments:

Reviewer Comment 1: This is a timely mixed-methods secondary analysis examining whether COVID-19–related stress and financial worry were associated with changes in opioid, cigarette, and alcohol use among people with HIV (PWH) with prior or current injection drug use in St. Petersburg, Russia, using data collected between May 2020 and July 2021. Key strengths include the focus on an under-studied, high-risk population outside the U.S. and the use of both survey and interview data to contextualize findings. With revisions—particularly clarifying measurement, outcome specification, statistical robustness, and interpretation—the paper could make a solid contribution.

RESPONSE: Thank you for your thoughtful review of our manuscript. We have made substantial revisions based on your feedback. Line numbers refer to the clean copy version.

Introduction

Reviewer Comment 2: The introduction effectively highlights inconsistencies in prior findings. It would benefit from clearer justification of why PLWH are a particularly relevant population for studying pandemic stress and substance use, and how the Russian context may differ from other regions in shaping these experiences. Expanding on these points would better articulate the study’s contribution and research gap.

RESPONSE: We have added additional justification in the introduction on the relevance of PLWH and the Russian context in studying pandemic stress and substance use. Thank you for this suggestion.

Changes in manuscript: The additional justification has been added to the introduction on lines 113-122 and 138-152.

Methods, Results, and Discussion

Reviewer Comment 3: Some details from the parent trials that are not directly analyzed here could be streamlined to avoid confusion (e.g., ART status, COVID infection history). The inclusion/exclusion table is useful, but the text should focus more tightly on variables relevant to this analysis.

RESPONSE: We have removed details about the parent studies that are not relevant to the present analysis.

Changes in manuscript: Removal of this information is on lines 173-181 and lines 213-215.

Reviewer Comment 4: The definition of “no change” combines “not changed,” “never used,” and “stopped using before the pandemic,” while “changed” includes increased, decreased, or stopped use due to the pandemic. Including never-users and pre-pandemic stoppers in the reference group may dilute or bias associations, particularly for alcohol and cigarettes, and creates inconsistencies across substances. If reanalysis is not feasible, a sensitivity analysis excluding these groups—or clearer discussion of this limitation—is recommended.

RESPONSE: Thank you for this recommendation. We have reanalyzed the data per your suggestion. For each substance (opioids, cigarettes, alcohol), we reran the same adjusted main model but restricted the analytic sample to participants with pre-pandemic use of that substance. This approach removes participants that responded with “never used” or “stopped using before the pandemic.” This has impacted the results in tables 3-5 and the corresponding text in the methods and results. We also made additions to the descriptive table (Table 2), for more transparency on how the variables were constructed.

Changes in manuscript: Updates to the description of the variable are located on lines 236-250. Updates to the results are located in the abstract, Tables 2-5, and text on lines 314-383.

Reviewer Comment 5: Causal language should be tightened. This is a cross-sectional analysis with stress and perceived change measured at the same time point, so directionality cannot be established. The manuscript should more clearly frame findings as associations and avoid implying causation.

RESPONSE: Thank you for pointing this out. We have updated language throughout the manuscript to frame the findings as associations and removed causal-implying language.

Changes in manuscript: Language updates occur throughout the manuscript, including pages have been made in lines 377-380, 513-515, 523-531, and 571-582.

Reviewer Comment 6: Dichotomizing stress and financial worry at ≥4 on a 1–5 scale requires stronger justification or sensitivity analyses. Reporting correlations between stress and financial worry, or considering them jointly in models, would strengthen interpretation.

RESPONSE: We thank the reviewer for this suggestion. COVID-related stress and financial worry were moderately correlated when continuous (ρ=0.61) or dichotomized (Spearman ρ=0.41), indicating related but distinct constructs. We added this information in the results section. We conducted sensitivity models including both variables simultaneously, and we found that estimates were attenuated, likely reflecting shared variance and limited power. Importantly, results were consistent when modeling both variables as continuous predictors, with both stress and financial worry significantly associated with opioid use change. These findings support the robustness of our results and suggest a graded relationship rather than a strict threshold effect. These additional models are represented in tables 6 and 7 as sensitivity analyses.

Changes in manuscript: Correlation values are reported on lines 321-323. Additional sensitivity models reported in Tables 6 and 7, and lines 384-390.

Reviewer Comment 7: Interpretation of the main finding could be clearer. The outcome reflects “any change,” not direction, and most changes were decreases. Reframing conclusions accordingly would better align with the data. Language describing financial worry as “clinically meaningful” should be softened given imprecision in the estimate.

RESPONSE: Thank you for this helpful comment. We have revised the Results and Discussion to more clearly reflect that the outcome represents any change in substance use rather than direction of change. Specifically, we now consistently describe findings as associations with “reporting any change” in opioid use. We also explicitly note in both the Results and Discussion that the majority of reported changes were decreases rather than increases, to better align interpretation with the observed data.

In addition, we have softened our interpretation of the association with financial worry. We have removed the phrase “clinically meaningful” and now describe this estimate as suggestive but imprecise, noting that the confidence interval includes the null.

Changes in manuscript: Language updates occur throughout the manuscript, including in lines 374-383, 509-516, 523-531, 572-582, and 610-618.

Reviewer Comment 8: The mixed-methods approach appears more explanatory than truly convergent, as qualitative findings are used primarily for contextualization. Clarifying the design and noting reliance on English-language memos (rather than full transcript coding by the integrative analyst) as a limitation would improve transparency.

RESPONSE: We appreciate the reviewer’s comment regarding the explanatory role of qualitative data. While qualitative findings do provide important context for interpreting quantitative results, this study is best characterized as a convergent mixed-methods design, as both data types were collected concurrently and neither informed the collection of the other. We have revised the manuscript to reflect this terminology and added a citation to established mixed-methods frameworks. We have also added the memo process as a limitation.

Fetters MD, Curry LA, Creswell JW. Achieving integration in mixed methods designs-principles and practices. Health Serv Res. 2013;48(6 Pt 2):2134-2156. doi:10.1111/1475-6773.12117

Changes in manuscript: We have added a s

---

## [Decision Letter · Decision Letter 1]

3 May 2026

COVID-19 pandemic-related stress and substance use behaviors among people with HIV – a mixed method analysis

PONE-D-25-55857R1

Dear Dr Rossi,

We’re pleased to inform you that your manuscript has been judged scientifically suitable for publication and will be formally accepted for publication once it meets all outstanding technical requirements.

Within one week, you’ll receive an email detailing the required amendments. When these have been addressed, you’ll receive a formal acceptance letter, and your manuscript will be scheduled for publication.

An invoice will be generated when your article is formally accepted. Please note that if your institution has a publishing partnership with PLOS and your article meets the relevant criteria, all or part of your publication costs will be covered. Please make sure your user information is up to date by logging in to Editorial Manager® and clicking the ‘Update My Information' link at the top of the page. For questions related to billing, please contact billing support.

Kind regards,

Vipula Rasanga Bataduwaarachchi, MD

Academic Editor

PLOS One

Reviewers' comments:

Reviewer's Responses to Questions

**Comments to the Author**

1. If the authors have adequately addressed your comments raised in a previous round of review and you feel that this manuscript is now acceptable for publication, you may indicate that here to bypass the “Comments to the Author” section, enter your conflict of interest statement in the “Confidential to Editor” section, and submit your "Accept" recommendation.

Reviewer #2: All comments have been addressed

2. Is the manuscript technically sound, and do the data support the conclusions?

Reviewer #2: Yes

3. Has the statistical analysis been performed appropriately and rigorously? 

Reviewer #2: I Don't Know

4. Have the authors made all data underlying the findings in their manuscript fully available?

The PLOS Data policy requires authors to make all data underlying the findings described in their manuscript fully available without restriction, with rare exception (please refer to the Data Availability Statement in the manuscript PDF file). The data should be provided as part of the manuscript or its supporting information, or deposited to a public repository. For example, in addition to summary statistics, the data points behind means, medians and variance measures should be available. If there are restrictions on publicly sharing data—e.g., participant privacy or use of third-party data—they must be specified.

Reviewer #2: Yes

5. Is the manuscript presented in an intelligible fashion and written in standard English?

Reviewer #2: Yes

6. Review Comments to the Author

Reviewer #2: Greetings

The authors have addressed all comments and revised the manuscript accordingly; it is now suitable for publication.

Kind regards,

7. PLOS authors have the option to publish the peer review history of their article (what does this mean?). If published, this will include your full peer review and any attached files.

Reviewer #2: No

---

## [Editor Report · Acceptance letter]

PONE-D-25-55857R1

PLOS One

Dear Dr. Rossi,

I'm pleased to inform you that your manuscript has been deemed suitable for publication in PLOS One. Congratulations! Your manuscript is now being handed over to our production team.

Kind regards,

on behalf of

Dr. Vipula Rasanga Bataduwaarachchi

Academic Editor

PLOS One